# Enhanced Ambient Sensing Environment—A New Method for Calibrating Low-Cost Gas Sensors

**DOI:** 10.3390/s22197238

**Published:** 2022-09-24

**Authors:** Hugo Savill Russell, Louise Bøge Frederickson, Szymon Kwiatkowski, Ana Paula Mendes Emygdio, Prashant Kumar, Johan Albrecht Schmidt, Ole Hertel, Matthew Stanley Johnson

**Affiliations:** 1Department of Environmental Science, Aarhus University, DK-4000 Roskilde, Denmark; 2Danish Big Data Centre for Environment and Health (BERTHA), Aarhus University, DK-4000 Roskilde, Denmark; 3AirLabs, Nannasgade 28, DK-2200 Copenhagen N, Denmark; 4Global Center for Clean Air Research (GCARE), School of Sustainability, Civil and Environmental Engineering, Faculty of Engineering and Physical Sciences, University of Surrey, Surrey GU2 7XH, UK; 5Department of Ecoscience, Aarhus University, DK-4000 Roskilde, Denmark; 6Department of Chemistry, University of Copenhagen, DK-2100 Copenhagen Ø, Denmark

**Keywords:** low-cost sensors, metal oxide sensor, electrochemical sensor, calibration protocol, calibration

## Abstract

Accurate calibration of low-cost gas sensors is, at present, a time consuming and difficult process. Laboratory calibration and field calibration methods are currently used, but laboratory calibration is generally discounted due to poor transferability, and field methods requiring several weeks are standard. The Enhanced Ambient Sensing Environment (EASE) method described in this article, is a hybrid of the two, combining the advantages of a laboratory calibration with the increased accuracy of a field calibration. It involves calibrating sensors inside a duct, drawing in ambient air with similar properties to the site where the sensors will operate, but with the added feature of being able to artificially increases or decrease pollutant levels, thus condensing the calibration period required. Calibration of both metal-oxide (MOx) and electrochemical (EC) gas sensors for the measurement of NO_2_ and O_3_ (0–120 ppb) were conducted in EASE, laboratory and field environments, and validated in field environments. The EC sensors performed marginally better than MOx sensors for NO_2_ measurement and sensor performance was similar for O_3_ measurement, but the EC sensor nodes had less node inter-node variability and were more robust. For both gasses and sensor types the EASE calibration outperformed the laboratory calibration, and performed similarly to or better than the field calibration, whilst requiring a fraction of the time.

## 1. Introduction

Poor air quality (AQ) constitutes a global public health emergency. Estimates of the global death toll caused by air pollution reach up to 9 million per year, or ∼1 in 6 deaths, and this number is increasing [1,2,3,4]. The impact of poor AQ on the cardiopulmonary system has been known for many years, but emerging studies link pollution exposure to a massive range of adverse health impacts, extending from dementia, Parkinson’s disease, and cognitive impairment, to diabetes, obesity and issues with the reproductive system [2,5,6,7]. It may, in fact, be damaging every organ in the human body [8]. Nitrogen dioxide (NO_2_) and Ozone (O_3_) are gas phase pollutants, with high spatiotemporal variability in urban environments and known health impacts [9,10]. The World Health Organisation Global Air Quality Guideline for NO_2_ is 10 μg m^−3^ (∼5 ppb) as an annual mean, and for O_3_ is 60 μg m^−3^ (∼30 ppb) as an 8-h mean for peak season [9].

Further efforts to quantify the links between AQ and health, and to produce targeted solutions, particularly for individual pollutants, are hampered however by the scarcity of AQ monitoring [11,12]. Reference standard monitoring utilises large and expensive instruments based on chemiluminescence for NO_2_ and UV photometry for O_3_ measurements [13,14], while pollutant concentrations vary rapidly in time and space, their measurement has low spatiotemporal resolution in developed countries due to the size and cost of traditional reference standard monitoring stations [12], and can be almost non-existent in low and middle-income countries, where the greater burden of poor AQ is felt [2,4,10]. Modelling of air pollution levels, through parameterised semi-empirical models, is an alternative, but often challenged by lack of high quality input and calibrated data, as modelling can not stand alone, but should be used in combination with measurements (see Hertel et al., 2007 [15]).

Low-cost sensors (LCS) have the potential to revolutionise air quality monitoring. There is no agreed definition of a low-cost sensor, however, they are far cheaper, smaller and record with greater time-resolution than traditional methods [12,16,17,18]. This means that they can be deployed in greater numbers, used in mobile applications, and in areas where monitoring is not currently possible, all with high time-resolution. Hence hot-spots, point-sources, indoor concentrations and even personal exposure levels can be identified and measured [12]. Monitoring programmes and epidemiological studies could also be implemented in lower income countries, which until now have had to reply on findings extrapolated from studies in other areas, that may underestimate the effects of poor AQ [19].

Many companies are already producing LCS commercially, and this has led to their use in scientific studies and citizen science projects, as well as privately [20,21]. However, together with the many benefits LCS have over traditional methods, they also have outstanding issues with data quality. In particular, LCS suffer from issues with selectivity, sensitivity, and stability, all of which detract from their overall accuracy [12,17,22,23]. These issues are typical to all LCS but also depend on the sensor’s principle of operation, and the context they are used in. In this study, we focus on gas sensors for NO_2_ and O_3_, based on Metal-Oxides (MOx) and Electrochemical (EC) cells. These are the most widely used LCS for NO_2_ and O_3_ measurement, with EC being more common [22,24]. They are both chemo-resistive sensors; MOx operate by measuring resistance change across a metal-oxide surface resulting from gas adsorption [25], and EC cells by using amperometry to measure the current of a redox reaction which is proportional to the gas concentration present in the air above the cell [26]. Further details of the individual sensors are given in Section 2. Both of these sensor types suffer particular issues with drift/ageing, cross-sensitivity between pollutants, and effects from temperature (T) [27,28,29].

Resultant data quality can be drastically improved with effective sensor calibration (and hardware approaches described in Section 2). Pre-deployment calibration is used to identify sources of error and develop calibration models that bring the sensors into best possible agreement with reference instruments [18]. Post-deployment calibration is necessary to maintain this during longer deployments, however, there is not currently a recognised standard procedure for calibration, and many commercial sensors are sold un-calibrated [22,30]. Calibration is individual and must be repeated for each LCS unit. Ideally each unit should also have a different calibration model for individual environments, as different co-pollutant levels and environmental factors will alter their response [24]. Once deployed, the drift in response will also be individual, dependent on both the LCS’s original characteristics and it’s environment. Therefore, individual re-calibration should also be performed when necessary [24,27].

The prevailing calibration methods are either Field (also called *co-location*) calibration or laboratory (Lab) calibration, both of which have known drawbacks [18,22]. Laboratory calibration can be performed relatively quickly, in any laboratory with appropriate equipment, and the range of pollutant concentrations, relative humidity (RH), and T levels, can be chosen. However, laboratory-based calibrations rarely perform well when validated in real field conditions [16,31] and, in a recent review of LCS for AQ monitoring, 90% of LCS studies surveyed (for NO_2_ and O_3_ measurement) utilised Field calibration [22]. The exact reason for this difference in efficacy has not been determined, however, it is thought to be due to meteorological and co-pollutant fluctuations in the field environment, which are not accurately represented in the laboratory, as well as the high cost associated with an effective laboratory calibration setup. There may also be contaminants in the laboratory environment for example volatile organic compounds (VOCs) and their oxidation products, not found in the field.

Field calibration is considered superior, particularly if calibration occurs in a similar geographical area to the actual measurement and in the same season of the year. However, it is time intensive, typically requiring several weeks to observe a comprehensive range of concentrations, particularly for more complex models, requires access to an official reference station or similar staging area, and leaves to chance whether or not the full range of pollutant concentrations is encountered [30,32,33,34]. Field calibration models may also not be transferable if moved between sites with different concentration profiles, co-pollutant matrices or prevailing meteorological conditions [22,35,36]. It has been claimed that the Laboratory and Field calibration methods are complementary and a combination of data from both methods is required for a full assessment of sensor performance and the production of a robust calibration model [16,26].

Within the different methods used to obtain calibration data (e.g., Field calibration and Laboratory calibration) there are many calibration models available, chiefly; Linear Regression (LR), Multivariate Linear Regression (MLR), or a range of Machine Learning (ML) algorithms, such as artificial neural networks, random forest and support vector regression, amongst others [22,30,36]. The most suitable calibration approach for all situations has not been determined. Complex ML algorithms often perform better than MLR when observing training data and if sensors are not moved after calibration, but can perform poorly after sensors are transferred to a different site [30,36]. This may be due to the complex models over-fitting specific aspects of the training site that are not directly related to pollutant concentrations, whereas, a simpler LR or MLR model may appear worse during training but not have a significant increase in error after transfer [33,37].

In this study, a ‘hybrid’ calibration method is described. This calibration method is a combination of both the Laboratory and the Field calibration methods, as the Hybrid calibration method draws ambient air into an insulated, well-mixed duct with a steady flow. This ensures that the air maintains similar properties to those observed outdoors. Inlets for adding NO_2_ and O_3_, as well as an activated carbon filter were also added to the duct. Thereby, sensor nodes placed inside the duct are exposed to ambient pollution levels, as well as artificially increased and lowered levels. The responses from the LCS are compared to reference monitors sampling the duct from the outside. This system is called the Enhanced Ambient Sensing Environment or EASE.

Previous works have been performed on the development of optimal laboratory calibration setups, but without the addition of ambient pollutants/conditions, this work was drawn on in the initial stages of the EASE design [38,39]. However, these typically focus on the manipulation and control of experimental conditions, whereas the main focus of the EASE setup is on preserving ambient conditions, whilst having the added advantage of manipulating the pollutant concentrations. This means that the LCS can be calibrated according to ambient concentrations, under realistic conditions in terms of RH, T, and the presence of a co-pollutant matrix, but with additional spiking of pollutants to ensure that the full span of pollutant concentrations is included in the calibration, within a short time-period. At points, filtration of the incoming air is also used, for identifying the zero/baseline response of the sensors under ambient conditions.

This study paves the way for future calibration methods that are faster and more accurate for real-world use. In this work, the EASE method is compared with Field and Laboratory calibration methods for 12 MOx and 15 EC sensors. Calibration was performed separately for the two sensor types and with differing setups. The main differences in the overall protocol is that the MOx sensors were EASE calibrated in Copenhagen, Denmark, and also Field calibrated and validated in Copenhagen, 3 km away, whereas, the EC sensors were calibrated in Copenhagen but were Field calibrated and validated in Surrey, UK, and with a longer separation in time. Furthermore, the MOx nodes were laboratory calibrated at Copenhagen University whereas the EC sensors were laboratory calibrated by their manufacturer prior to node assembly.

The main aim of the work is to compare the three different calibration methods (Field, Lab and EASE) for both sensor types (MOx and EC) as well as comparison between the sensors. The EASE method gave better results than Laboratory calibration and had similar results to Field calibration, whilst requiring a fraction of the time. The EC sensors performed better than the MOx nodes for NO_2_ measurement and similarly for O_3_ measurement, but with less inter-variability between nodes.

## 2. Materials and Methods

In this section, the sensor node hardware is described, followed by the calibration setups and procedures for each of the three calibration methods, separately, for both node types. The data analysis was conducted in R [40], using the ggplot2 [41] and Openair [42] packages for data visualisation.

### 2.1. Sensor Nodes

In this article, ‘sensor’ refers to individual sensing devices, e.g., a metal-oxide chip, whereas ‘sensor node’ or ‘node’ refers to a complete package including sensors, housing, sampling system, and the ability to internally log or broadcast data.

Two types of sensor nodes were used in this study, both are prototypes developed by AirLabs ApS for monitoring urban air quality and called ‘AirNodes’. Both nodes measure NO_2_ and O_3_, Generation 2 (Gen 2) AirNodes do so with MOx sensors, whereas Generation 5 (Gen 5) AirNodes utilise electrochemical cells. The relevant sensors to this study are detailled for both nodes in Table 1, schematics and images of the nodes are shown in Figure 1 and Figure 2.

#### 2.1.1. Gen 2 MOx Nodes

In the MOx node, the gas sensors used are MiCS-6814 metal-oxide sensors from SGX Sensortech [43]. Each sensor chip contains three sensing elements ($s_{1}$, $s_{2}$, $s_{3}$), each of these has a heating element, a magnified image of the sensor is shown in Figure A5. Sensing element three (pure WO_3_) is optimised for measuring oxidising gasses and therefore its output is primarily used in this context. Element 2 is designed for the measurement of reducing gases and element 1 for ammonia. The sensing mechanism of a MOx is reliant on surface reactions, therefore the grain size, thickness and porosity of the MOx surface layer will alter the sensitivity and response rate of the MOx sensor to a pollutant. These effects are explained further in other studies [44,45,46].

The method used to create the MOx film will affect the microstructure of it’s surface and therefore the performance, and even when the same method is used for sensor chips of the same model. the microstructure of each chip can vary. In use, the performance of individual sensor chips can be highly variable, meriting individual calibration. An image of the chips, showing surface structure, difference in drop size and damage to a sensing element is shown in Figure A5.

The chips are operated with temperature cycling operation (TCO), which is a technique developed by Schütze and coworkers [47]. Cycling the operating temperature means that at a certain point in the cycle the optimal temperature for binding of a specific gas will be reached, thus providing the highest sensitivity for the gas of interest. Species on the surface can also be burnt off at high temperatures, cleaning the surface. In the Gen 2 MOx sensor, the cycle is optimised for NO_2_ and O_3_ and the sensor output is recorded at the high and low points of each cycle. A schematic of the nodes is shown in Figure 1, two sensors (MOX1 and MOX2) measure the sampled air in series, with an integrated O_3_ filter between them, this removes O_3_ and therefore sensor 1 is exposed to both gasses, whereas sensor 2 (after the filter) is exposed to NO_2_ without the presence of O_3_. The output of both sensors is then used in determining the concentrations of the gasses, and the cross-sensitivity can be mitigated [48].

Each complete Gen 2 node is installed in a weatherproof enclosure (88 × 88 × 90 mm) with inlets, exhaust holes, and a fan for active sampling, illustrated in Figure 1. Active sampling was found to be integral to sensor performance. The sensing elements consume the gas of interest when measuring; if the air around the sensor were stagnant, a lower and less linear response would be observed than with active flow.

#### 2.1.2. Gen 5 EC Nodes

The Gen 5 nodes (shown in in Figure 2) contain EC cells from Alphasense, a NO2-B43F cell [49], which has a MnO_2_ filter that reduces O_3_, and a OX-B431 cell [50] which does not have the filter and is sensitive to both NO_2_ and O_3_. The O_3_ concentration can be found from the difference between cell responses (analogously to the MOx nodes). The EC cells do not rapidly consume the target gas and therefore active sampling is not included in the node design (schematics of the node are shown in Figure 2). The cells are sensitive to temperatures above 20 °C, however, and a sun shield is added to the node body to mitigate heat build-up. A temperature sensor is also included within the node and during this study the nodes internal temperature did not exceed the threshold of 20 °C at which a temperature correction is necessary. The node body dimensions are 190 × 105 × 70 mm.

Use of these sensors has been extensively reported in a number of studies [22,26,28], but in simple terms, each cell contains a working electrode (WE) and an auxiliary electrode (AE). The WE is exposed to the environment and is where the redox reactions occur, resulting in a change in current. The AE has the same structure as the WE but is not exposed to ambient air, and so is not affected by gas concentrations, only other environmental parameters such as temperature. The difference in output between the two electrodes, therefore, corresponds to changes in concentration at the EC cell surface. AirLabs have developed a printed circuit board (PCB) for converting the cell output from nA into mV, the board has a conversion rate of 0.735 mV/nA.

### 2.2. Calibration Method Overview

The MOx and EC nodes underwent separate calibration procedures, the timings of these are detailed in Table A4. In order to validate the methods, the nodes were co-located for several weeks at reference stations, this co-location was split into training for the Field calibration and a validation period. The same nodes were also calibrated over ∼3 days each in the EASE and Lab setups. The performance of the Field, EASE and Laboratory models over the validation period were then compared, a schematic of this is shown in Figure 3. The evaluation statistics for validation of the calibration models (for all methods) are calculated based on a comparison between the nodes and reference instruments during the Field validation period, with the different calibration models applied.

A general overview of the methods is presented in Table 2, and the protocols for each node and calibration method are described below.

#### 2.2.1. MOx Laboratory Calibration Protocol

Laboratory calibration of the MOx nodes took place inside a 1 m^3^ chamber, made from aluminium, stainless steel and Perspex, situated inside a larger climate-controlled chamber (Viessmann A/S), this setup is already partially described in Bulot et al. [51] (2020). A schematic of the Laboratory calibration setup is shown in Figure 4, it includes an ozone generator and O_2_ flask (pure O_2_), as well as an NO_2_ flask (1–2.3% NO_2_ in N_2_) and mass flow controllers (MFCs, 0–100 ml min^−1^). A Model 42i chemiluminescence NOx Analyser (Thermo-Fisher Scientific, Waltham, MA, USA) was used for NO_2_ measurement as well as a direct absorbance analyser, a Model 405 nm (2B Technologies, Boulder, CO, USA). Ozone was measured with a BMT 930 UV photometer, and RH and T were monitored with an HTU21D digital sensor. In order to control RH levels, filtered, dry air was supplied to the chamber, via a MFC directly (lowering RH) or diverted through a Nafion membrane submerged in water (increasing RH). The chamber air was mixed with three fans in X, Y and Z directions, test mixing with CO_2_ is shown in Figure A4.

The Laboratory testing protocol consisted of arranging the nodes inside the chamber and sealing it before steadily increasing pollutant concentrations (NO_2_ or O_3_ separately) from zero to ∼80 ppb, before allowing the concentration to steadily decay, at RH of 25, 50 and 75% (±15%), and T of 12 and 20 °C (±2 °C), respectively. Resulting in 6 concentration spikes for each pollutant in total, over the course of 3 days. An example time-series of of an O_3_ concentration spike is shown in Figure 5.

#### 2.2.2. MOx EASE Calibration Protocol

A schematic of the EASE calibration setup is shown in Figure 6. It consists of an insulated galvanised steel duct (32 × 32 cm cross-section), which acts as the mixing chamber, and is connected to a blower fan with 12 cm diameter ducting. The inlet to the system was extended 2 m away from the building through a window on the 5th floor. Inlets for the addition of NO_2_ (50 ppm NO_2_ in N_2_ flask) and O_3_ (pure O_2_ passed through an O_3_ generator) were also added to the main inlet and their flow controlled by MFCs (0–100 mL min^−1^). The duct is equipped with flow, RH and T probes (LS control ES991 and ES989), mixing fans, and outlets for NO_2_ and O_3_ monitors (Model 405 nm NOx monitor and BMT 930 UV photometer). As the setup is insulated, and a relatively large throughput of air is used, the interior of the chamber is similar to the ambient air outdoors, however the setup has the advantage of being able to artificially increase (with gas flasks or generators) or decrease (with an activated carbon filter) pollutant concentrations. An example section of a concentration profile, with spikes and a ‘rush-hour’ period, is shown in Figure 7.

The chosen EASE calibration protocol required 48 h per pollutant (or 72 h total for both pollutants as the ambient measurements can be used for NO_2_ and $O_{3}$ calibration), during which time the sensors were exposed to ambient pollutant concentrations throughout, except for zeroing periods and pollutant spikes, at low (∼40 ppb), medium (∼80 pbb) and high (∼100 pbb) concentrations for NO_2_ and O_3_ independently. With each lasting ∼45 min and occurring during non-rush hour periods. During the rush hour periods NO_2_ and O_3_ concentrations are not altered. The flow was held constant throughout and RH and T dictated by the ambient environment. A 24 h example section is shown in Figure 7.

#### 2.2.3. MOx Field Calibration Protocol

The MOx Field calibration took place at the H. C. Andersens Boulevard (HCAB) roadside monitoring station in Copenhagen, Denmark, operated under the NOVANA program (the Danish National Monitoring Program for Water and Nature [52]). HCAB is a highly trafficked street with relatively high air pollution levels. The nodes were co-located at the station from 22nd December 2020 until 3rd February 2021 (43 days). The period before 12th January 2021 was designated as the training period (21 days) and post as the validation period (22 days). During the co-location mean RH was 80% (range: 58–96%), and mean T was 2.1 °C (range: −6.2–9.1 °C). A time-series of the reference NO_2_ and O_3_ concentrations during the co-location is shown in Figure 8. The nodes were installed at the same height and approximately the same distance from the traffic at HCAB as the inlet for the reference instruments. A schematic of the setup is also shown in Figure 8.

The reference instrument for NO_2_ is a chemiluminescence NOx monitor (Teledyne API model T200). Reference O_3_ measurements were recorded via UV-absorption at 254 nm with a Teledyne API model T400. The detection limit for both monitors is below 1 ppb, with precision of ±5%. Reference station data is recorded at 30 min time-resolution and therefore the LCS data was averaged to the same level for analysis.

#### 2.2.4. MOx Calibration Models

As the Laboratory and EASE models in this study must inherently be transferable between sites, a simple MLR was chosen to produce the calibration models, to avoid overfitting to a specific environment [30,36]. An individual stepwise-selected model was used for each of the MOx nodes for all of the MOx calibration methods, where the input variables were the following, for predicting both NO_2_ and O_3_:(1)$$\begin{array}{cc} \; & P_{NO_{2}\;\mathrm{or}\;O_{3}}=\\ & a_{0}+a_{1}*MOX1\_s_{2}\_high+a_{2}*MOX1\_s_{2}\_low+a_{3}*MOX1\_s_{2}\_offset+a_{4}*MOX1\_s_{2}\_scale+\\ \; & a_{5}*MOX1\_s_{3}\_high+a_{6}*MOX1\_s_{3}\_low+a_{7}*MOX1\_s_{3}\_offset+a_{8}*MOX1\_s_{3}\_scale+\\ \; & a_{9}*MOX2\_s_{2}\_high+a_{10}*MOX2\_s_{2}\_low+a_{11}*MOX2\_s_{2}\_offset+a_{12}*MOX2\_s_{2}\_scale+\\ \; & a_{13}*MOX2\_s_{3}\_high+a_{14}*MOX2\_s_{3}\_low+a_{15}*MOX2\_s_{3}\_offset+a_{16}*MOX2\_s_{3}\_scale+\\ \; & a_{17}*RH+a_{18}*T \end{array}$$
where $a_{0}$ is the offset and $a_{1}$–$a_{16}$ are the calibration coefficients, and $a_{17}$ and $a_{18}$ are the temperature and relative humidity correction coefficients, calculated using the method of multiple least squares, separately for each MOx node. Due to the temperature cycle (described in Section 2.1.1), the MOx sensor provides several parameters obtained as raw data, as seen in Equation (Equation 1). For each sensor chip ($s_{2}$ and $s_{3}$), a scale and offset are provided for both the high- and low-temperature period in the temperature cycle, and the average conductance during the two temperature periods is also provided. Since two sensors are included in the MOx node, and measure the air in series, with an O_3_ filter between them, the readings from both sensors (MOX1 and MOX2) are included. Parameters from sensing element two ($s_{2}$) and three ($s_{3}$) are both included as input variables in the model, even though only $s_{3}$ is optimised for oxidising gases. Parameter $s_{2}$ was included to check for interference, but if none were found (*p*-value > 0.05), the input variables were removed before doing the stepwise-selected model. The stepwise-selected model is determined based on the step function in R with the mode of stepwise search done with a ‘forward’ and ‘backward’ direction until the lowest Akaike Information Criterion (AIC) was found. Temperature and RH were included as input parameters in the models, but the stepwise-selection model disregarded them as inputs as they did not improve the final MOx node calibration models.

#### 2.2.5. EC Laboratory Calibration Protocol

The EC cells (NO_2_-B43F [49] and OX-B431 [50]) are produced by Alphasense and each cell is tested in a single pass setup, yielding zero current (nA) and sensitivity (nA ppm^−1^) values, prior to dispatch [53]. Tests are conducted with a flow of 5 L min^−1^, T of 22±2 °C and RH of 45±15%. Alphasense is confident in the linearity of cell response within the specified measurement range (0–20 ppm) and therefore only test at pollutant concentrations of zero ppb, and one known concentration (not disclosed). Tables of sensitivities and zero values for the cells are included in the Appendix A (Table A1 and Table A2).

#### 2.2.6. EC EASE Calibration Protocol

For the EC nodes, a scaled-up duct system was constructed (38 × 50 cm cross-section) to accommodate the larger node bodies, and different gas monitors were used, a Thermo Electron Model 42C chemiluminescence NOx Analyzer and a Thermo Scientific Model 49i Ozone Analyzer. Otherwise, the calibration followed a similar procedure to the MOx EASE calibration, except that ∼120 ppb was used for the ‘high’ exposure level, and the inlet concentration was filtered during the artificial pollutant spikes for better consistency. An example plot of the full procedure is displayed in Figure 9. The EASE setup in Figure 6 is also representative of the EC calibration setup.

#### 2.2.7. EC Field Calibration Protocol

The EC co-location took place at Surrey University, Guildford, United Kingdom, where the EC nodes were mounted in front of the Thomas Telford Building Air Quality Lab, on the university campus, at ∼1 m from the ground. This is the same level as the intake of the reference monitors (Figure 10), which are operated by Surrey University. The co-location took place between 9th December 2021 and 23rd December 2021 (15 days), and was split into training and validation periods which consist of measurements before and after 15th December 2021, respectively. During the co-location internal node mean RH was 44% (range: 32–58%) and internal node mean T was 11.2 °C (range: 1.10–18.1 °C). Since the co-location took place on the university campus, low NO_2_ levels were encountered, relative to the HCAB co-location. The O_3_ levels were similar, a time-series of reference pollutant concentrations is shown in Figure 10. The reference instrument for NO_2_ was a Serinus Ecotech 40 NOx monitor, whereas reference O_3_ measurements were recorded via a Thermo Fisher Scientific 49i Ozone monitor. All reference data was obtained with 1-mintime resolution, and therefore the node data did not need to be aggregated.

#### 2.2.8. EC Calibration Models

For the Laboratory calibration, the supplied coefficients (see Table A1 and Table A2 in the Appendix A) and the recommended equations are utilised. Concentrations of NO_2_ are predicted from the NO2-B43F cell (cell 2), using Equation (Equation 2) to find corrected WE output ($WE2_{C}$), from raw WE output ($WE2_{v}$), WE sensor zero ($WE2_{0}$), temperature dependent correction factor ($n_{T_{1}}$), raw AE output ($AE2_{v}$) and AE sensor zero ($AE2_{0}$). Which is then used in Equation (Equation 3) with the supplied sensitivity ($S_{2}$) and an offset ($C_{2}$) to convert the output into ppb. The sensitivities, sensor zeroes and temperature dependent correction factor are supplied by the manufacturer, the offset (if included) is determined by Field co-location. The sensitivities are multiplied by −10 due to the change in cell output when using the AirLabs PCB and not an Alphasense PCB, this is omitted from the equations for clarity.
(2)$$WE2_{C}=\left(WE2_{v}-WE2_{0}\right)-n_{T_{1}}\left(AE2_{v}-AE2_{0}\right)$$
(3)$$P_{NO_{2}}=WE2_{C}*S_{2}+C_{2}$$

Predicted O_3_ concentration is calculated from the difference in response between the two cells. Corrected output for cell 1 WE ($WE1_{C}$) is determined in the same way as $WE2_{C}$, except that the variables relate to cell 1, as shown in Equation (Equation 4), and afterwards, the O_3_ concentration can be determined based on Equation (Equation 5): (4)$$WE1_{C}=\left(\left(WE1_{v}-WE1_{0}\right)-n_{T_{2}}\left(AE1_{v}-AE1_{0}\right)\right)$$
(5)$$P_{O_{3}}=\left(WE1_{C}*S_{1}-WE2_{C}*S_{2}\right)+C_{1}$$

Alterations were made to these stock equations after testing. The supplied temperature dependent correction factor ($n_{T_{1}}$) for NO_2_ prediction was 1 if T is 0–10 °C and 0.6 for temperatures 10–20 °C, whereas $n_{T_{2}}$ for O_3_ prediction was 1.5 if T is below 10 °C and 1.7 for temperatures 10–20 °C, although when this was applied, the $R^{2}$ value of the model dropped drastically. When the difference between modelled and reference NO_2_ or O_3_ concentration was plotted against temperature for the modelled concentration without a temperature correction, there was no relation between the error and temperature in the range experienced at the test site, and therefore the $n_{T_{1}}$ and $n_{T_{2}}$ values were set to 1. This is in agreement with other deployments of the sensors, where a temperature dependence is not noticeable below 20 °C.

For the EASE and Field calibration models, similar equations are used, however the supplied sensitivities and zero values for the cells were not included. For prediction of NO_2_ concentrations the voltage change in cell 2 is found from the difference between the WE and AE outputs, as shown in Equation (Equation 6), and the predicted O_3_ concentration is found by subtracting cell 2 response from cell 1 response, as shown in Equation (Equation 7).
(6)$$P_{NO_{2}}=\left(WE2_{v}-AE2_{v}\right)*S_{N}+C_{N}$$
(7)$$P_{O_{3}}=\left(\left(WE1_{V}-AE1_{V}\right)-\left(WE2_{V}-AE2_{V}\right)\right)*S_{O}+C_{O}$$

The sensitivities, $S_{N}$ and $S_{O}$, as well as offsets, $C_{N}$ and $C_{O}$, are determined using the method of multiple least squares, separately for each EC node.

## 3. Results and Discussion

In the following section, the different calibration methods are compared, firstly for the MOx nodes, followed by the EC nodes, and finally the different sensor types are compared with each other. Results from the validation of the EASE setup against a reference station are included in the Appendix A, Section A.1.

In the review by Karagulian et al. (2019) [22], a good level of agreement for a sensor with a reference instrument is denoted by a $R^{2}$ value of >0.75 and a slope ‘close’ to 1.0, which we take to mean 1±0.3, this definition will be used in the following analysis. The $R^{2}$ value gives a measure of the goodness of fit between variables but does not account for bias. Relative bias is denoted by a slope that diverges from 1. A non-zero intercept denotes absolute bias and impacts the limit-of-detection, this is also recorded and discussed for each method [22]. The root mean square error (RMSE), mean bias error (MBE), and mean absolute error (MAE) are also included as statistical indicators for the models.

### 3.1. Mox Node Results

Results for the validation of MOx sensor calibration with each method are displayed in Table 3 and selected statistical indicators are shown in Figure 11. Example time-series of NO_2_ and O_3_ for one of the nodes during the validation period are shown in Figure 12 and Figure 13, respectively.

It was found that individual models of this prototype node were highly variable and some were unstable and gave poor results. Nodes that had training $R^{2}$ values < 0.75 and or validation $R^{2}$ < 0.1 were discounted from the study, consequently, 5 of the 12 nodes had to be discounted. The poor reliability of the nodes is presumably due to issues with the complex sampling/filtration system, or the fragile MOx chips, as some of the broken nodes had low sampling flows or damaged chips, an example chip is shown in Figure A5). After calibration, it was found that the response of the MOx cells changed dramatically at temperatures <0 °C, presumably due to perturbation of the TCO. Negative temperatures were present only in the Field validation period and not the Field training (or EASE or Laboratory training), and all models performed poorly when temperatures were <0 °C in the validation period, therefore, these data were also removed from the validation to better compare the models.

Overall, MOx sensor measurement of O_3_ was superior to NO_2_, with $R^{2}$ values of 0.96 vs. 0.83 in the Field validation, and better performance for all calibration methods. In terms of the different calibration methods, the Laboratory method was least successful, followed by the EASE method, with the Field being most effective. All six statistical indicators follow this pattern (Laboratory < EASE < Field) in terms of $R^{2}$ and slope difference from 1, intercept difference from zero, and size of RMSE, MBE, and MAE, except for Field NO_2_ MBE, which is greater than EASE MBE, and slope which is the same for all methods. In the case of slope and intercept, a mean value can be misleading as these parameters can be above or below the optimum value (1 or 0, respectively), which is why the standard deviations and boxplots are included. One of the nodes calibrated for NO_2_ with the Laboratory method had a negative slope (−0.9) for the validation, this pulled the mean slope value closer to 1 despite being a poor and anomalous result and therefore was removed from the analysis. Despite the EASE and Field methods performing better than the Laboratory method, they do not meet the requirements for ‘good’ sensor performance for NO_2_ measurements, due to slope values that are too high (1.6). However, the EASE and Field calibrated O_3_ measurements are well within the requirements, with particularly high $R^{2}$ values (0.93 and 0.96) and acceptable slope values (1.2 and 0.88).

In conclusion, the EASE method outperforms the Laboratory method with similar overall results to the Field method, but the sensor hardware only performs well enough for O_3_ measurement, as even the Field calibrated NO_2_ measurements are not within the requirements for a good sensor, under these conditions. The MOx nodes also had poor reliability, with a large fraction being discounted, and did not perform well at negative temperatures (although they were not trained under negative temperatures).

### 3.2. EC Node Results

Results for validation of the EC sensors with all three methods are displayed in Table 4, and selected statistics are displayed in Figure 14. Example time-series of NO_2_ and O_3_ during the validation period are shown in Figure 15 and Figure 16, respectively. Results from the calibration training periods are found in Table A6 in the Appendix A. The mean NO_2_ concentration was low during the entire period (training and validation) at 10.4 ppb, and particularly low during the training period (7.7 ppb, vs. 13 ppb in the validation period). This means that at times the levels are near the limit of detection for the cells (reported previously as ∼4 ppb [54]). Therefore a lower $R^{2}$ value is recorded for the sensor output compared with the reference instruments in the training period, relative to the validation period (0.49 vs. 0.83), whereas the mean O_3_ concentrations were greater throughout, and the $R^{2}$ for the O_3_ measurement is similar for both periods (∼0.83) [55,56]. The Field calibration performs very well for O_3_ ($R^{2}$ = 0.83, slope = 0.97) but over-predicts NO_2_ concentrations in the validation period (slope = 1.4), which is most likely due to the short training period, with lower NO_2_ levels than the validation period. As stated previously, ∼3 weeks or longer is recommended for Field calibration.

The $R^{2}$ and RMSE values for validation of the EC nodes with different methods are essentially the same, unlike the MOx nodes for which $R^{2}$ differs between methods. This is because the EC cells have a more linear response to pollutant concentrations, and less output variables, and therefore a change in the slope between methods does not alter the coefficient of determination. This is also reflected by a generally better performance and lower inter-unit variability for the EC nodes.

It was found that the the Laboratory coefficients provided by the EC manufacturer (Alphasense) yield results that scale well with concentration increase in the Field, with a slope similar to the Field calibrations (∼1.3), however the intercept of the Laboratory predicted concentrations were either greatly above or below zero (between −175 and +126 ppb) when applied to the Field data. Consequently, a Field offset correction was also identified, based on the the Field calibration training period. The offsets are shown in the Appendix A in Table A3 and as an example by the light blue lines in Figure 15 and Figure 16. This means that the pure Laboratory calibration would only be usable for measuring relative changes, and not absolute values, unless a short Field co-location is performed to determine their offset (as is the case for the statistics in Table 4 and the dark blue lines in Figure 15 and Figure 16).

The zero coefficients supplied by Alphasense were included in the Laboratory calibration models, as described in Section 2.2.8, but this only partially reduced the range of intercepts encountered for the validation period (e.g., from 184 ppb down to 166 ppb for NO_2_), compared with not using the zero coefficients, and just subtracting the AE from the WE response. The offset issue may be partly due to the use of a PCB designed by AirLabs in the Gen 5 EC node, which complicates the use of the ’zero’ coefficients supplied by Alphasense, as the electronic offset may be altered. However, the large variability in offset between units suggests that individual offset calibration is necessary for each cell in the Field, regardless of the PCB used. In the application note from Alphasense, it is stated that large over/under-estimation may occur if using just Laboratory coefficients and that a secondary correction method is normally required [53], which was the case here. However, the scale for the models produced by the Laboratory calibration was impressive. Potentially a simple on-site zero air calibration, or calibration at a single fixed concentration, could be used together with the Laboratory calibration model coefficients to improve their results. This will be tested as part of a future study.

Temperature correction using the $n_{T}$ coefficients supplied was also tested but did not improve the results. A correlation linking the difference between model predictions and reference measurements was not observed for NO_2_ measurements during the Field validation. For O_3_, a decrease in absolute difference was observed for increasing temperatures, meaning that the sensors appear to perform better for O_3_ measurements at greater temperatures, however O_3_ concentration also had a strong correlation with temperature, meaning that the increased absolute accuracy is due to the sensors having lower absolute error at greater concentrations. When temperature was included in the training of the MLR model for the Field method it did not improve results in the Field validation. This suggests that below 20 °C using the difference between the WE and AE is sufficient to account for any sensitivity that the EC sensors have towards temperature increase. This is also evidenced by the strong performance from the Field trained model ($R^{2}$ = 0.83, slope = 0.97), which does not include temperature.

In conclusion, the Field and EASE methods perform similarly for both NO_2_ and O_3_ measurements, all being in good agreement with the reference, except for Field NO_2_ which has a slope slightly above the acceptable range, as described above. It is also clear to see from the time-series in Figure 15 and Figure 16 that these methods result in models that track the observed data well. In this case, EASE performs essentially as well as the Field calibration, with a better slope for NO_2_, worse for O_3_, but meeting the criteria for both. This is despite the fact that EASE calibration was performed in CPH and the nodes validated in Surrey, UK, although it should also be noted that the Field calibration was shorter than is optimal. Overall we believe this study demonstrates EASE as a viable alternative to Field calibration for the EC nodes.

### 3.3. Node Comparison

As can be seen in Figure 8 and Figure 10, the concentration profiles in the Field co-locations periods are very different. The HCAB co-location (MOx nodes) is longer and importantly for the NO_2_ calibration has a large NO_2_ concentration range during the Field training period (0–81 ppb). Meanwhile the Surrey Field training period (EC nodes) is short and has a lower mean NO_2_ level (7.7 ppb), with a lower NO_2_ concentration range (0.50–29 ppb), followed by a validation period with a slightly greater mean NO_2_ concentration (13 ppb). A lower mean concentration can result in a lower $R^{2}$ value regardless of the sensor being tested, and validating sensors using a concentration range greater than the training range is not optimal [34,55,56]. Half-hourly data is the maximum resolution available from HCAB, whereas the Surrey co-location data had a one-minute time resolution, using greater time-resolution generally results in lower performance statistics. Therefore, it is difficult to compare the Field calibration of the sensors under these differing conditions. However, the EC nodes appear to have been trained and validated under more challenging conditions and yet still perform as well as or better than the MOx nodes.

Temperature is also a factor in the training/validation of the models, but again is difficult to compare. It was warmer during the Surrey co-location than the HCAB co-location (11 vs. 2.8 °C), but the sensors respond differently to temperature changes. The MOx sensors operate poorly below 0 °C, whereas the EC cells operate well at negative temperatures, but lose sensitivity at greater temperatures (particularly above 20 °C), which were not present during the co-location.

The laboratory MOx training consisted of a number of concentration, T and RH combinations (although temperatures were greater than those encountered in the field). In contrast, the EC Laboratory calibration consisted of a zero calibration and one other concentration/T/RH combination. This may partly explain the large offsets produced by the EC Laboratory method. In terms of the EASE training, the MOx nodes were trained and tested in the same geographical area, which is the preferred method for EASE calibration, whereas the EC nodes were trained in Copenhagen, Denmark and tested in Surrey, UK. Despite this, the EASE method performs well for the EC nodes.

It is clear that the EC Gen 5 nodes are more robust and less variable than the MOx nodes as none of them had to be discounted from the analysis, compared with 41% of the Gen 2 MOx nodes. It seems that the fan/filter system in the MOx nodes and sensors chips themselves are vulnerable to damage.

Overall, If using the best calibration method in each case (Field or EASE), the statistics for the different sensors appear similar, for NO_2_ measurement, EC: $R^{2}$ = 0.83, slope = 1.2, MOx: $R^{2}$ = 0.83, slope = 1.6 and for O_3_ measurement, EC: $R^{2}$ = 0.83, slope = 0.97, MOx: $R^{2}$ = 0.96, slope = 0.88. However, when all factors are taken into account, including the less optimal EC Field co-location and the unreliability of the MOx nodes, the EC nodes are judged as superior.

## 4. Conclusions

This study demonstrates how the EASE calibration method performs better than pure Laboratory calibration and similarly to a Field calibration (and in some cases better, e.g., EC NO_2_), whilst requiring a fraction of the time, being completed in days instead of weeks. The EASE method even performed well when nodes were calibrated in Copenhagen, Denmark and validated in Surrey, UK, up to 3 months later, suggesting that using a site with similar characteristics (e.g., Urban, European) and at least within the same season, yields acceptable results. Although, we expect even better results from calibration at or nearer to the intended measurement site and directly before deployment.

Although the EASE method performed well under the circumstances in this study and can expose sensors to the full expected range of pollutant concentrations in a condensed period, it does not ’condense’ the RH and T exposure. These meteorological parameters have less impact on the node output than concentration range, at least within certain ranges (e.g., T < 20 °C for EC and >0 °C for MOx), therefore in most cases, we expect that if the calibration is performed in the same season as the deployment an EASE calibration will be sufficient. However, the temperature during the co-location should be monitored and if it is outside of the optimal operating range for the sensors, then either a separate temperature correction should be applied, or data may need to be discounted. The EASE method will be best applied to shorter deployments, e.g., one season, but is still expected to perform better than Laboratory calibration over longer deployments. The issue with not capturing an appropriate range for meteorological factors is also present for Field calibration methods, as can be seen from the MOx data in this study. Theoretically, longer Field calibrations could solve this issue, for instance a year long co-location would ensure the sensors encounter a large meteorological range, but this is not viable for most LCS, and in particular for EC sensors, due to their overall lifetime and the drift they exhibit during deployment. We propose that additional EASE calibration periods during a deployment would be a better solution. The speed of an EASE calibration also means it is a viable option for conducting pre- and post-deployment calibration, and using the the combined calibration models to account for drift.

Laboratory calibration did not produce a calibration model meeting the requirements for a ‘good’ sensor performance in any of the cases in this study. A potential improvement would be to model the expected concentration, RH and T of the site during a deployment and then to use a range around them for the defined Laboratory parameters. Although this would not account for potential co-pollutant species present at the deployment site.

The secondary purpose of this study was to compare the Gen 2 MOx nodes and the Gen 5 EC nodes. It is clear from the results that despite the Field co-location being less optimal for the EC nodes (shorter, lower concentration range, lower mean NO_2_ level, warmer), they out-perform the MOx nodes for NO_2_ measurements, for O_3_ the results from both sensor types were similar and rate as ‘good’ for Field and EASE calibration. However, the MOx nodes were less reliable.

The method is in its infancy but we expect that further testing and iteration of the procedure will improve the results, particularly when dealing with correlated or anti-correlated pollutants that the sensors are cross-sensitive to. The natural progression for this work would be to either install an EASE system directly at a reference station and use that for optimal EASE calibration for the surrounding area, or to build a mobile EASE system inside a vehicle that could be used for condensed calibration at the exact site at which nodes will be measuring, and to validate the sensors in the same setup. This could provide rapid and accurate calibration on-demand. Based on the results from this study, the method appears to perform well enough to invest in this.

## Figures and Tables

**Figure 1 sensors-22-07238-f001:**
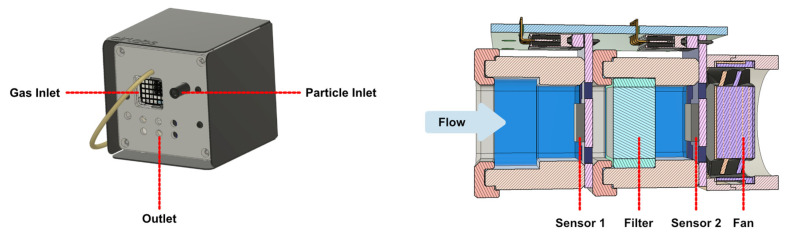
AirNode Gen 2 (**left**) and cross-section of the AirNode Gen 2 gas sampling system, showing both of the MOx cells in series, the filter between them, and the fan behind them (**right**).

**Figure 2 sensors-22-07238-f002:**
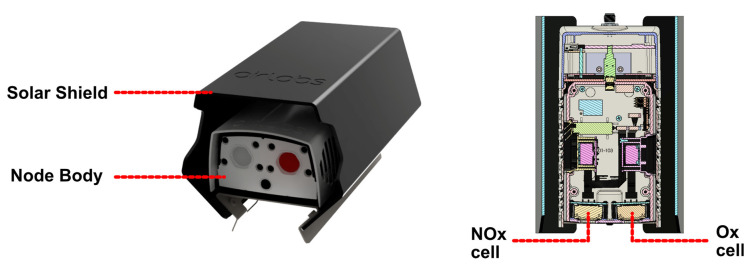
Gen 5 node views, node with heat-shield installed (**left**), and cross-section of the node with sensor locations (**right**).

**Figure 3 sensors-22-07238-f003:**
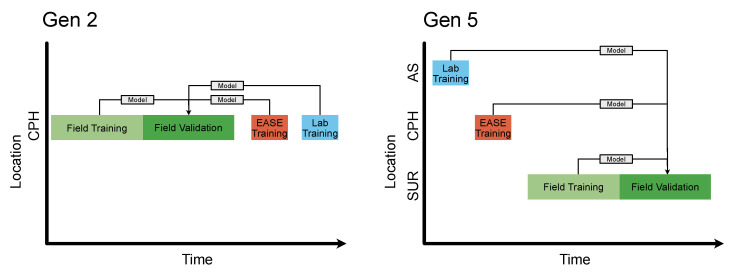
Schematic of calibration periods. The blocks sizes are not to scale for time, Field training should be 7 times the size of EASE and Laboratory training. Arrows indicate calibration models. The CPH location stands for Copenhagen, AS for Alphasense, and SUR for Surrey. Lab is short for laboratory.

**Figure 4 sensors-22-07238-f004:**
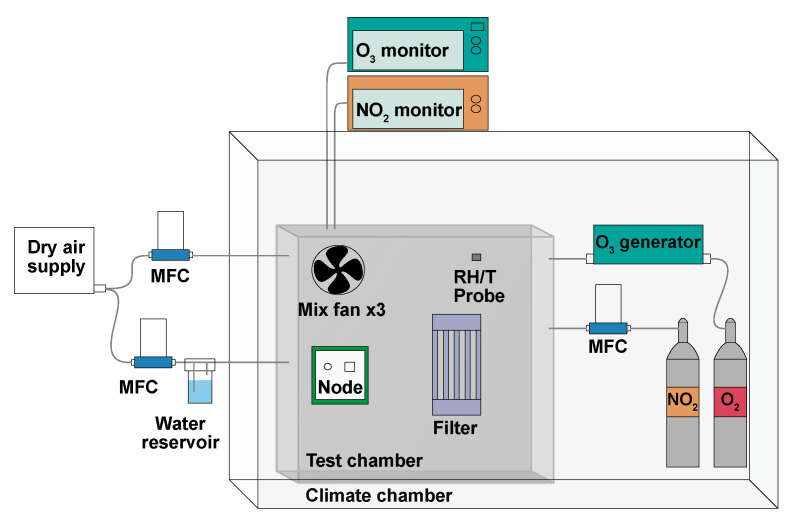
Laboratory calibration setup schematic for the Gen 2 MOx nodes.

**Figure 5 sensors-22-07238-f005:**
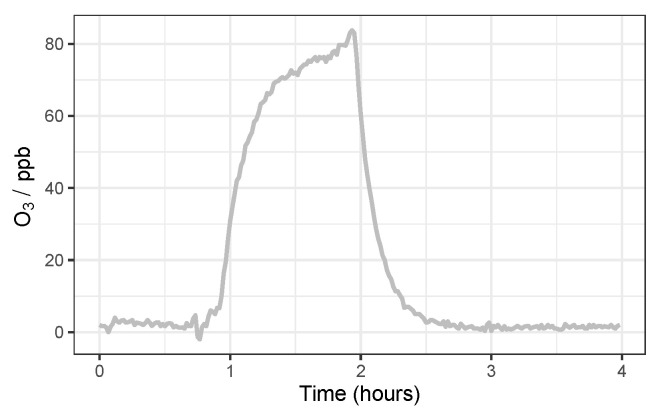
Example O_3_ spike from the MOx Laboratory calibration.

**Figure 6 sensors-22-07238-f006:**
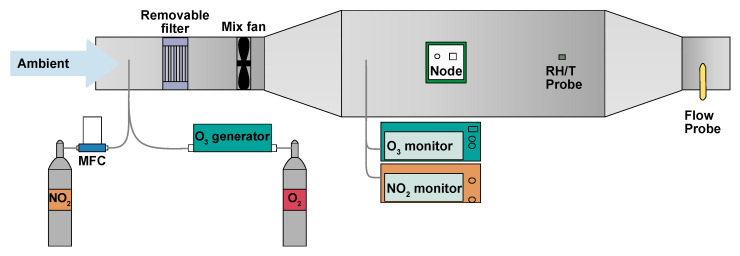
EASE calibration setup. This schematic is representative of the setup used for both MOx and EC calibration.

**Figure 7 sensors-22-07238-f007:**
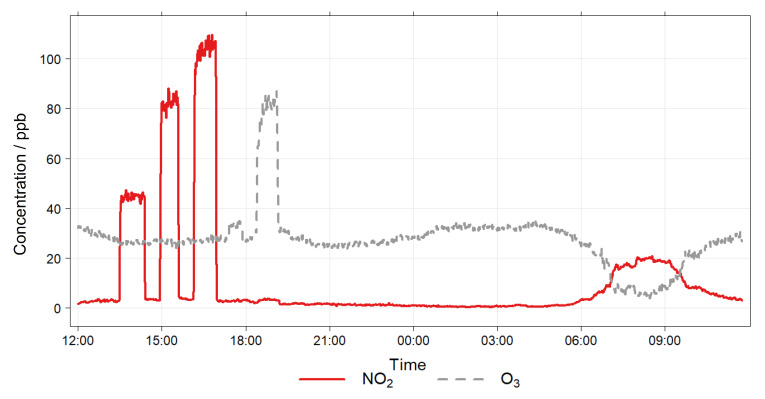
EASE calibration setup example concentrations, note the artificial spikes from 12:00 to 21:00, and morning rush hour period from 06:00 to 10:00.

**Figure 8 sensors-22-07238-f008:**
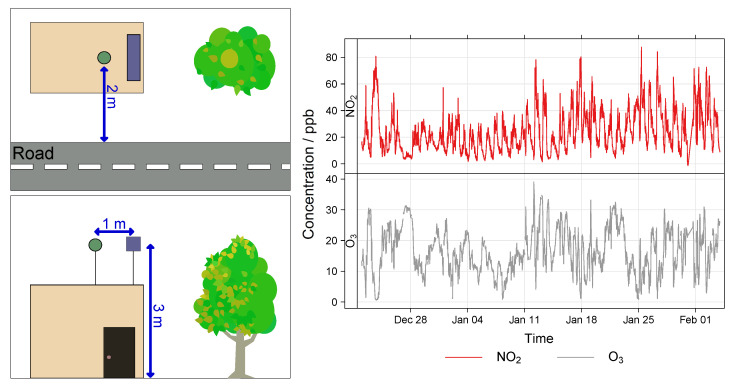
MOx Field calibration and validation setup (**left**) and pollutant time-series (**right**). In the setup schematic the node location is the blue rectangle and the reference inlet is the green circle.

**Figure 9 sensors-22-07238-f009:**
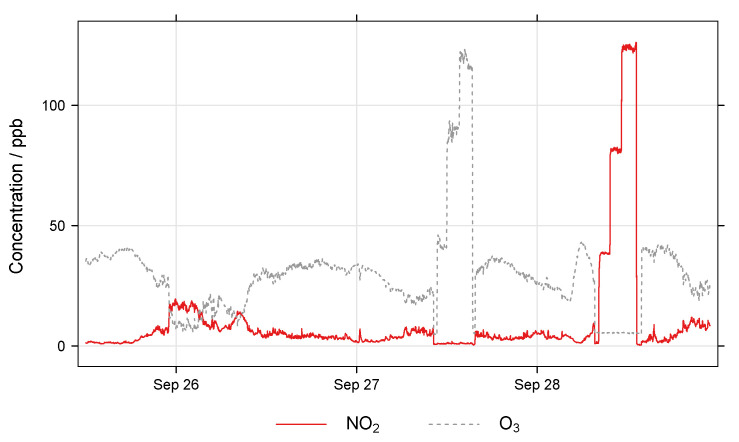
Example concentration profile from EC calibration in the EASE setup.

**Figure 10 sensors-22-07238-f010:**
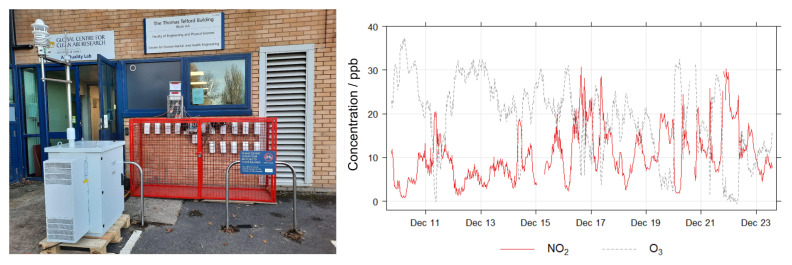
EC Field calibration and validation setup (**left**), and pollutant time-series (**right**).

**Figure 11 sensors-22-07238-f011:**
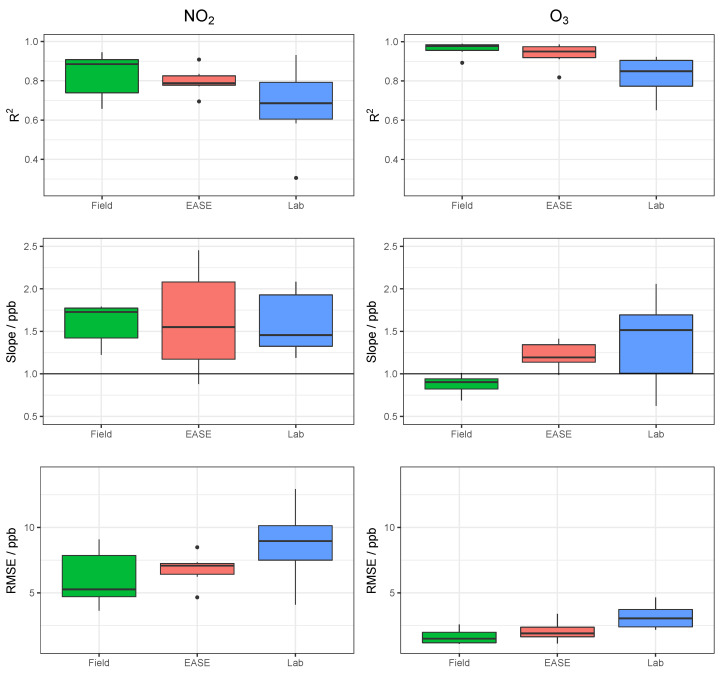
Comparison of $R^{2}$, slope, and RMSE for calibration model validation of MOx sensors with the different methods for both NO_2_ (**left**) and O_3_ (**right**). Lab is short for laboratory.

**Figure 12 sensors-22-07238-f012:**
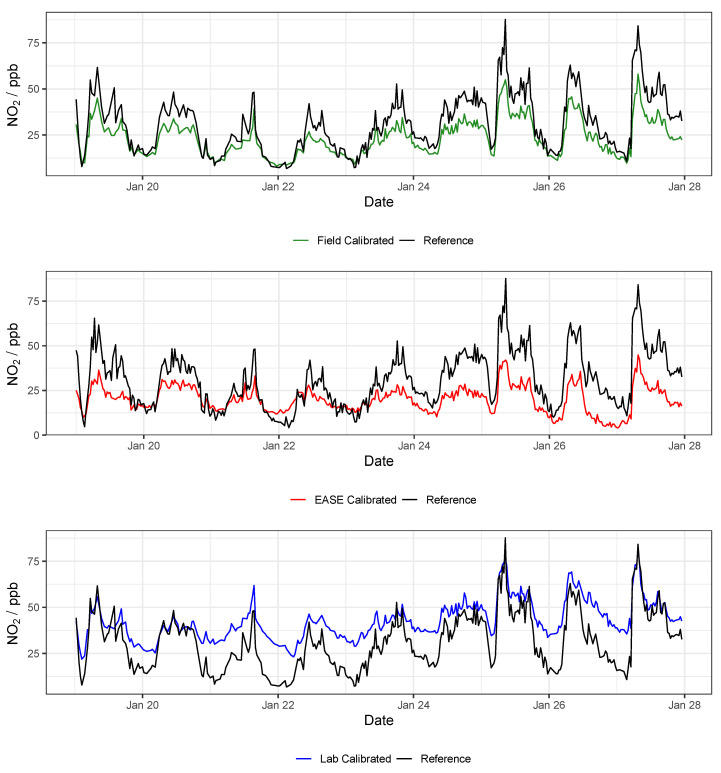
Example NO_2_ time-series from a single MOx node during the validation of the different calibration methods, compared with data from the reference instrument. Lab is short for laboratory.

**Figure 13 sensors-22-07238-f013:**
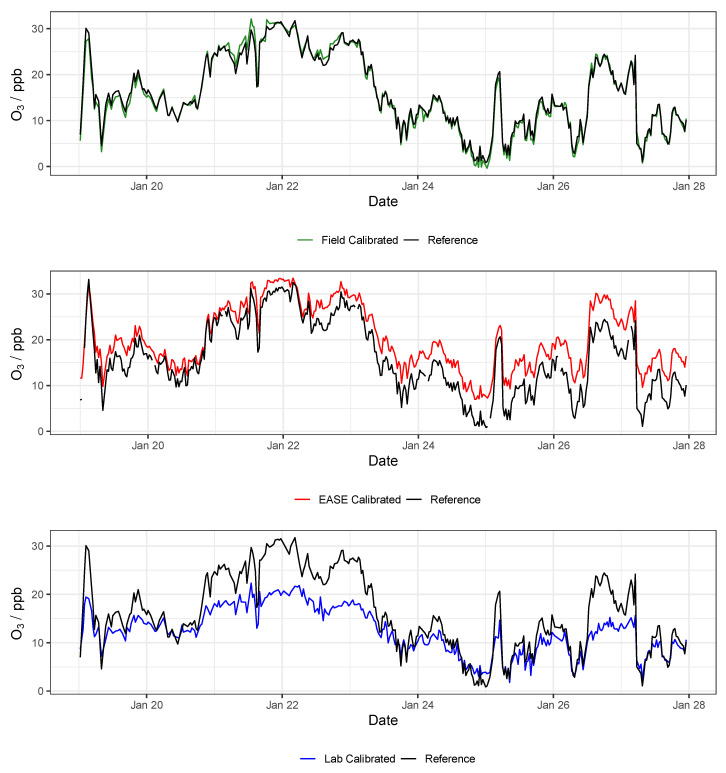
Example O_3_ time-series from a single MOx node during the validation of the different calibration methods, compared with data from the reference instrument. Lab is short for laboratory.

**Figure 14 sensors-22-07238-f014:**
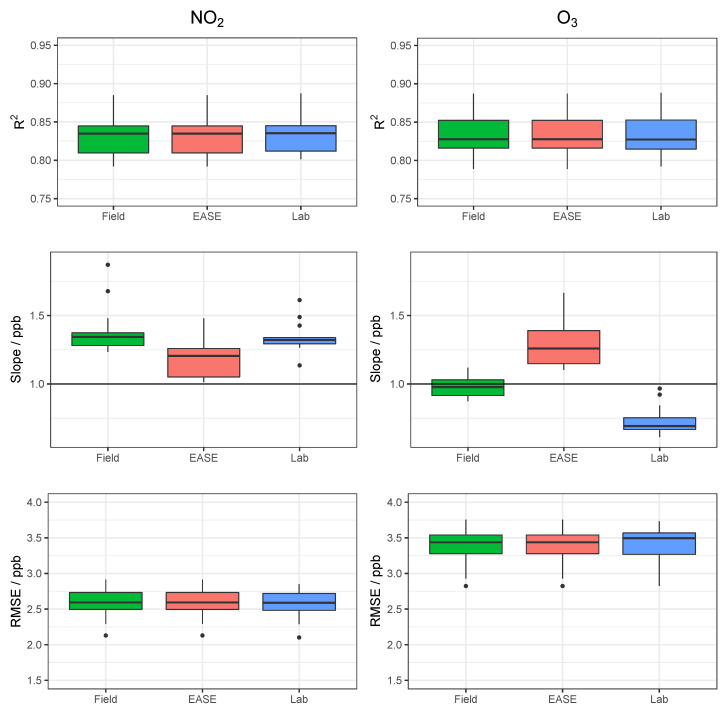
Comparison of $R^{2}$, slope, and RMSE for calibration model validation of EC sensors with the different methods for both NO_2_ (**left**) and O_3_ (**right**). Lab is short for laboratory.

**Figure 15 sensors-22-07238-f015:**
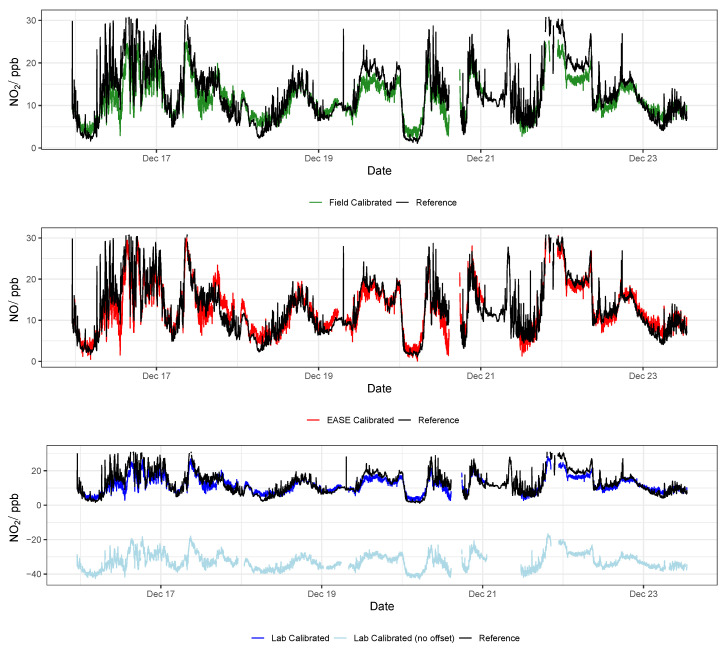
Example NO_2_ time-series from a single EC node during the validation of the different calibration methods, compared with data from the reference instrument. Lab is short for laboratory.

**Figure 16 sensors-22-07238-f016:**
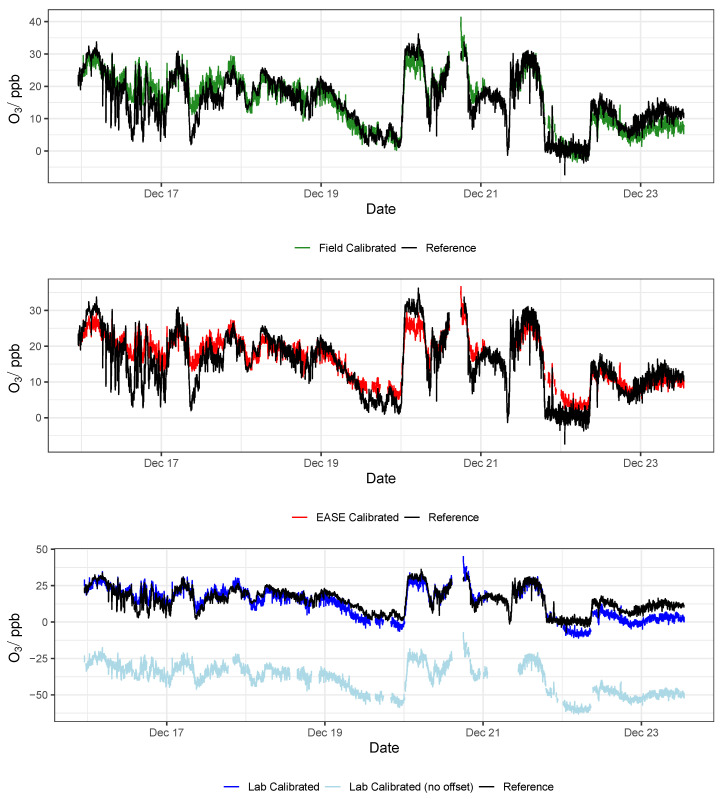
Example O_3_ time-series from a single EC node during the validation of the different calibration methods, compared with data from the reference instrument. Lab is short for laboratory.

**Table 1 sensors-22-07238-t001:** Relevant sensors within the Gen 2 and Gen 5 nodes. The output column describes sensor output after processing and calibration.

Node	Sensor	Producer	Type	Output
Gen 2	MiCS-6814	SGX Sensortech/AirLabs	MOx	NO_2_/ppb
Gen 2	MiCS-6814	SGX Sensortech/AirLabs	MOx	O_3_/ppb
Gen 5	NO2-B43F	Alphasense/AirLabs	EC	NO_2_/ppb
Gen 5	OX-B431	Alphasense/AirLabs	EC	O_3_/ppb

**Table 2 sensors-22-07238-t002:** Calibration method overview. A schematic showing the time-series for the different methods is included in Figure 3.

	Field	Laboratory	EASE
**RH/%**	Ambient	25, 50, 75	Ambient
**T/°C**	Ambient	10, 20	Ambient
**[C]/ppb**	Ambient	0-80-0	Ambient + 0, 40, 80
**Time taken**	∼3 weeks (preferably)	∼3 days	∼3 days
**Resource intensity**	Low (but requires station access)	High	Medium

**Table 3 sensors-22-07238-t003:** Evaluation statistics for validation of the MOx nodes with all of the calibration methods, all values are means over all MOx nodes with their corresponding standard deviation of the results shown in brackets.

Pollutant	Method	$R^{2}$	Slope	Intercept/ppb	RMSE/ppb	MBE/ppb	MAE/ppb
NO_2_	Laboratory	0.67 (0.22)	1.6 (0.39)	11 (21)	8.5 (3.0)	−16 (17)	21 (11)
	EASE	0.80 (0.065)	1.6 (0.63)	−11 (19)	6.8 (1.2)	−3.5 (13)	13 (2.7)
	Field	0.83 (0.12)	1.6 (0.24)	−6.5 (4.9)	6.2 (2.2)	−7.7 (3.2)	8.8 (2.9)
O_3_	Laboratory	0.82 (0.11)	1.4 (0.54)	−6.4 (5.3)	3.2 (0.98)	5.2 (16)	11 (12)
	EASE	0.93 (0.062)	1.2 (0.16)	−2.2 (6.9)	1.9 (0.90)	−1.3 (4.5)	4.3 (1.4)
	Field	0.96 (0.037)	0.88 (0.12)	1.4 (0.9)	1.4 (0.67)	0.87 (2.8)	2.2 (2.3)

**Table 4 sensors-22-07238-t004:** Evaluation statistics for validation of the EC nodes with all calibration methods. Note that the statistics for the Laboratory calibration are calculated after the Field offsets, shown in Table A3, have been applied. All values are means over all EC nodes with their corresponding standard deviation shown in brackets.

Pollutant	Method	$R^{2}$	Slope	Intercept/ppb	RMSE/ppb	MBE/ppb	MAE/ppb
NO_2_	Laboratory	0.83 (0.025)	1.3 (0.11)	−2.7 (1.1)	2.6 (0.20)	−1.1 (0.42)	2.4 (0.19)
	EASE	0.83 (0.027)	1.2 (0.13)	−2.6 (5.2)	2.6 (0.22)	0.32 (4.2)	3.9 (2.2)
	Field	0.83 (0.027)	1.4 (0.17)	−2.6 (1.4)	2.6 (0.22)	−1.6 (0.49)	2.6 (0.39)
O_3_	Laboratory	0.83 (0.027)	0.73 (0.10)	6.1 (2.3)	3.4 (0.27)	−2.8 (1.7)	4.5 (1.1)
	EASE	0.83 (0.027)	1.3 (0.17)	−8.3 (7.6)	3.4 (0.27)	2.7 (4.9)	5.3 (2.5)
	Field	0.83 (0.027)	0.97 (0.073)	−0.092 (1.7)	3.4 (0.27)	0.47 (0.95)	2.8 (0.29)

## Data Availability

All raw data is available upon request.

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
