# Peer review of "Enhanced Ambient Sensing Environment—A New Method for Calibrating Low-Cost Gas Sensors"

_sensors, 2022, doi:10.3390/s22197238_

Round 1
Reviewer 1 Report
The manuscript, titled with “Enhanced Ambient Sensing Environment (EASE) - A New Method for Calibrating Low-Cost Gas Sensors”, presents NO2 and O3 gas sensors based on metal-oxide (MOx) and electrochemical (EC) devices. The authors compare them in Lab, EASE and field conditions and try to present the advantages of EASE. However, the context, graphs and measurement are not organized. The calibration items in the target are also not revealed. There are a lot of information but not correlating to the topic. From the conclusion and results in Table 4 and 5, EASE is not important for the gas sensors.
1. Table 1: It is not expressed in the context for all information. Why is there so many sensors list here but only used two or three ones in the study? What does OPC mean? The significance for each part should be linked to the topic and explained in the contents. However, most of them only appears in the table, not showing characteristic features to this topic.
2. Figs. 1, and 2: The functions and the corresponding kits/parts should be revealed. Simply showing the drawings/photos here is not significant to this topic.
3. Fig. 3: What do the “CPH” and “SUR” mean? This figure cannot work for this paper because its contents and symbols are not well delivered in this paper. It’s too rough to use the curves/arrows to represent the calibration model.
4. Figs. 5, 7, 8 (right column), 9, 10 (right): Figs. 5, 7, 8 (right column), 9, 10 (right) are the test results and should be presented after the equations (1)~(6).
5. Figs. 4, 6 and 8 (left): They are different methods and should be discussed at the same section for each kit, part, and dimension. Are they both fit MOx and EC sensors?
6. Tables 2 and 3: It is hard to understand for Table 3. For Table 2, EASE should be emphasized and link to Figs. 3, 4 and 6 in the text.
7. Tables 4 and 5: What do the parameters mean? What is the factor relation to get the slope and R^2?
8. Figs. 12, 13, 15 and 16: What are the method and setup to get the reference curves?
9. Eqs. (1)~(6): Because they are provided from the sensor vendors, what is the effort from this study? The equations are very important to this calibration and they should be developed by this paper, not using from the device manuals.
I have read all the text but still cannot understand overall parameters and factors due to the paper organization and language quality. I suggest the authors to re-organize the paper as NO2 and O3 sensors with specific instruments/kits to perform better properties, not emphasizing the EASE.
10. The EASE is not important because of only adding tubes and filters to manipulate the gas samples. It is a normal idea to use EC and MOx sensors. The cross factors of humidity and temperature are also normal to the sensors. So that, the title and paper focus should be changed.
11. The appendix is too complicated and not useful to the readers to catch the paper points. Cancel it please.
Reviewer 2 Report
Please find the attachment for the comments.

Reviewer 3 Report
In the manuscript entitled “Enhanced Ambient Sensing Environment (EASE) – A New Method for Calibrating Low-Cost Gas Sensors”, H. S. Russell et al. have described the calibration of an enhanced ambient sensing environment method, comparing the response towards NO2 and O3 pollutant gases of low-cost sensors to references.
First of all, in the title and in the abstract the authors should remove acronyms.
In the manuscript the authors refer to lab instead of laboratory; please, correct.
In the abstract, the authors refer to “lab and field methods”; it is not clear, the authors should specify to what they refer to.
Please, substitute “veracity” with “accuracy”.
In general, the abstract is too long and poor comprehensive: reduce it, briefly indicating the motivation and the obtained objectives of the proposed study; moreover, also in the abstract the authors should specify which types of gases are detected and their concentration range.
The R2 data in the abstract are not clear, please report the obtained results in a more comprehensive way.
In the introduction, the authors should report the standard procedures used to monitor air pollution.
In the introduction, the authors should introduce the gas sensor technology, comparing it to the standard methodologies used to monitor air pollution; moreover, the authors should indicate which type of transduction sensing mechanisms are involved in commercial gas sensors.
Define and describe low-cost gas sensors.
In the introduction, the authors should indicate the common pollutant gases, their concentration range, and the common technology used to detect them.
The authors should briefly describe MOx based-gas sensor and electrochemical gas sensors, comparing them, reporting advantages and disadvantages.
In the introduction, the authors should define the sensor calibration procedure.
In the introduction, the authors should define “hybrid” calibration method.
Why have the authors chosen only NO2 and O3 gases? Discuss.
Which is their reported calibration method sensitivity, selectivity, repeatability and accuracy? Discuss.
The authors should indicate and motivate the choice of the place where they have placed the sensor system.
In my opinion, to confirm the effectiveness of their procedure, the authors should evaluate tests towards also other gaseous pollutants, at various concentration range, different mixture of them, considering also the environment conditions (e.g. temperature, relative humidity, etc.).
It is not clear if the purpose calibration method is effective in real conditions with the presence of various pollutant gases.
The English style should be improved.
I can accept this manuscript with major revisions.
Reviewer 4 Report
The work is devoted to Enhanced Ambient Sensing Environment (EASE) method, which is a hybrid of the two, combining the advantages of a lab calibration with the added veracity of a field calibration. The study was carried out to a good level using a wide range of research devices and models. The experimental results are described in detail. The manuscript is well written, with almost no typos. The manuscript may be published after a minor revision changes listed below.
Notes:
l.31. “Modelling is an alternative,…” - It should be briefly explained what kind of modeling - numerical, calibration, etc.
Table 1. “OPC” and “NDIR” abbreviations should be explained.
l.292 Perhaps "15-10-2021" should be written as "15-12-2021".
l.438-441. “Meanwhile the Surrey Field training period (EC nodes) is short and has a lower mean NO2 level (7.7 ppb), with a lower NO2 concentration range (0.50-29 ppb), followed by a training period with a slightly greater mean NO2 concentration (13 ppb). ". It is unclear expression. Perhaps the last part of the expression assumes "validating period”. So l.381-382 show another phrase.
In chapter 2 "calibration protocols" are well written. Perhaps a brief description/overview of the validation methodology should be added to improve comprehension.
Round 2
Reviewer 1 Report
The responded contents are not complete but I would like to release them to follow the other reviewers.
Reviewer 3 Report
In my opinion, the revised version of the manuscript can be accepted for publication.